# Bridging the Preference Gap: Post-Training Input Rewriting with Large Language Models

## Abstract

Pre-trained language models, such as BERT and RoBERTa, have achieved remarkable performance in semantic classification tasks. Yet, their effectiveness varies with different textual expressions due to inherent preferences developed during training. To address this limitation, we propose a framework that leverages large language models (LLMs) to rewrite input texts in ways that better align with a target classifier's preferences, thereby enhancing its performance. To achieve this, we introduce a training process for the LLM and an automated method for constructing training data that encapsulates the classifier-specific preferences. Furthermore, we present a multi-sampling and filtering strategy to address instability in LLM outputs. Empirical evaluations on semantic classification datasets demonstrate that our framework significantly improves classifier's performances.

## 1 Introduction

Semantic text classification (Altınel & Ganiz, 2018), such as classic sentiment classification or natural language inference (NLI), has long been a cornerstone of natural language processing. The field has witnessed transformative progress through the evolution of various approaches, ranging from traditional machine learning models to pre-trained compact architectures like BERT (Devlin, 2018) and large language models (LLMs) such as GPT-3 (Brown, 2020). These models leverage self-supervised pre-training and task-specific fine-tuning to achieve state-of-the-art performance. However, a fundamental characteristic persists across model scales: inherent preferences (Jia & Liang, 2017; Naik et al., 2018). Being data-driven, these models often exhibit varying performance across different textual expressions with identical semantics due to biases and patterns inherent in their training data.

Traditional approaches have focused on eliminating or mitigating model preferences through data augmentation techniques. These include lexical-level perturbations (e.g., synonym replacement, random token deletion), semantic-preserving transformations like back-translation (Sennrich, 2015), and generative methods employing GANs (Goodfellow et al., 2020) or LLMs to synthesize new examples. These methods operate in a pre-training paradigm, where the goal is to enhance model performance by generating more diverse training data before the model is fully trained. While such strategies diversify training data and superficially shift model preferences by exposing it to paraphrased inputs, they ultimately fail to eliminate preferences entirely and may even lead to suboptimal solutions. This fundamental limitation stems from the intrinsic nature of preferences (Jia & Liang, 2017; Naik et al., 2018) – they are ingrained characteristics of models shaped by their architectural biases and optimization trajectories, where even augmented data inevitably introduces new preference patterns. Instead of endlessly trying to remove these preferences, we believe it's more practical to work with them.

To realize this, we propose a post-training approach harnessing the linguistic capabilities of LLMs to dynamically rewrite input texts according to target model preferences during the inference phase. Unlike traditional methods that rely on pre-training data augmentation, our approach adapts the input to align with the model's inherent characteristics at inference time, offering two key advantages: (1) it eliminates the need for retraining the model and (2) it unlocks the model's full potential by aligning inputs with its preferred patterns. By aligning input expressions with a model's preferred patterns during inference, we aim to bridge the gap between arbitrary inputs and the model's optimal comprehension space. This approach introduces two core challenges: (1) How to identify preference-

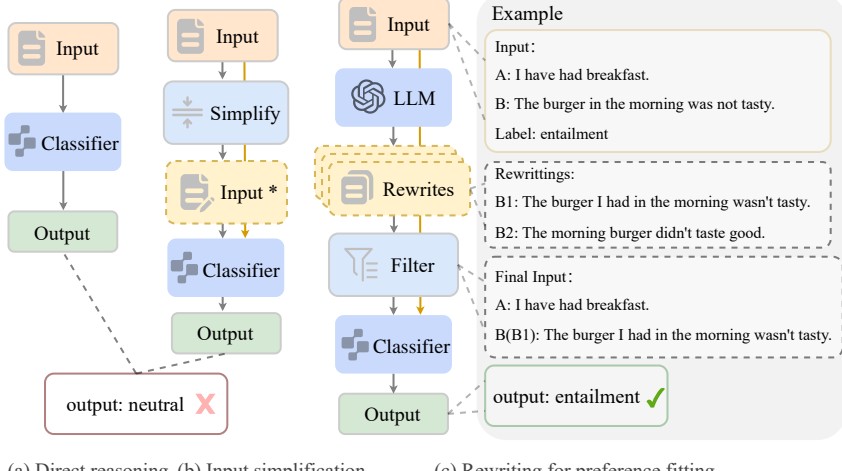

(a) Direct reasoning    (b) Input simplification      (c) Rewriting for preference fitting

Figure 1: Overview of our approach. From left to right, it shows using an Classifier for direct inference, methods for simple normalization of input, and our approach. The example is an NLI task where the goal is to determine whether the relationship between sentence A and sentence B is entailment, contradiction, or neutral.

aligned textual features that maximize model performance, and (2) How to ensure stable, high-quality rewrites from stochastic LLM generations.

To address the first challenge of identifying preference-sensitive features, a natural starting point is to hypothesize that simplifying textual complexity (e.g., shortening sentences, using high-frequency vocabulary, or simplifying syntax) could align inputs with model preferences. This intuition stems from extensive psycholinguistic studies on human language processing, where reduced linguistic complexity has been shown to lower cognitive load and improve comprehension (Spencer et al., 2019). However, as shown in Table 1, our empirical investigations reveal a striking divergence between human and model behaviors: classification accuracy remains remarkably stable across texts with widely varying complexity levels. Whether sentences contain 5 words or 25 words, whether vocabulary consists of common terms or technical jargon, or whether syntactic structures are flat or deeply nested—these complexity dimensions show negligible correlation with model performance. This finding fundamentally challenges the assumption that model preferences can be optimized through linguistically-driven complexity reduction.

To overcome the limitations of complexity-based approaches highlighted above, we develop a novel framework combining implicit preference learning with robust generation control. As shown in the Figure 1 First, we automatically construct a preference dataset capturing performance variations across semantic equivalents, then train an LLM through Direct Preference Optimization (DPO) (Rafailov et al., 2024) to internalize the target model's decision patterns. During inference, we implement a multi-sampling strategy with quality filtering—generating multiple candidate rewrites and selecting the most stable consensus choice through similarity clustering. This design directly addresses the second core challenge of ensuring reliable rewrites by mitigating the inherent stochasticity of LLM generations, contrasting with conventional single-sample decoding approaches that suffer from output instability.

To summarize, our contributions are as follows:

- We empirically validate that model preferences operate through mechanisms distinct from human linguistic cognition, demonstrating that traditional text complexity metrics (e.g., sentence length, lexical rarity) cannot reliably predict model behavior;
- We introduce a method for training the LLM to capture the preferences of the Classifier, along with a technique for the automated construction of training data;
- We present a method that combines multiple sampling with a filtering strategy to address the issue of instability in the outputs generated by the LLM.

Table 1: Experiments on the Impact of Text Complexity on Model Performance. We repeatedly sampled pairs of correctly and incorrectly predicted examples from the dataset. For each dataset, there are two data points: the data on the left represents the proportion of instances where correctly predicted examples scored higher than incorrect ones on the respective metric, while the data on the right represents the opposite. The four metrics are: text length, average word frequency of all words in the sample, average number of dependencies, and number of distinct part-of-speech tags.

| Metric | Datasets | | | | | | | | | | |
|---|---|---|---|---|---|---|---|---|---|---|---|
| | MRPC | | MNLI | | QQP | | RTE | | QNLI | | SST2 |
| Text length | 55.8 | 43.6 | 46.6 | 53.4 | 55.8 | 43.6 | 48.7 | 51.3 | 49.8 | 49.9 | 44.8 | 54.5 |
| Word frequency | 54.1 | 45.9 | 51.1 | 48.9 | 48.6 | 51.4 | 55.0 | 45.0 | 49.2 | 50.8 | 50.5 | 49.3 |
| Dependencies | 54.8 | 45.2 | 50.1 | 49.6 | 52.0 | 47.9 | 48.1 | 51.9 | 49.8 | 50.0 | 52.7 | 47.0 |
| Diversity | 47.0 | 39.9 | 41.4 | 41.2 | 42.0 | 39.9 | 42.3 | 37.5 | 37.5 | 42.6 | 44.4 | 42.1 |

Experimental results indicate that our method effectively enhances performance across various subsets of the GLUE benchmark when applied to three different Classifiers. Specifically, it improves the performance of the BART-base model by 0.72 points and the RoBERTa-large model by 1.08 points on the overall GLUE dataset.

## 2 RELATED WORKS

### 2.1 PSYCHOLINGUISTIC STUDIES

The role of linguistic complexity in cognitive and psycholinguistic research has gained significant attention. Studies have demonstrated that textual complexity, encompassing lexical, syntactic, and discourse-level features, significantly influences human cognitive processing and judgment (Spencer et al., 2019) in tasks such as text comprehension, readability assessment, and information recall, as evidenced by neuroimaging studies showing differential brain activation patterns in response to varying levels of linguistic complexity (Ferstl et al., 2008; Pylkkänen, 2019). At the lexical level, metrics such as word frequency, word length, and lexical diversity (Kyle et al., 2018) have been extensively examined in relation to human cognitive load and processing efficiency. Inspired by these findings in human cognitive processing of linguistic complexity, we aim to investigate whether similar linguistic features affect NLP model performance, potentially enabling us to optimize text generation for better model comprehension and task performance.

### 2.2 PREFERENCES MODELING

In earlier studies on model preference, the focus was primarily on experiments assessing models' robustness to different expressions. For instance, adversarial examples have been used to evaluate reading comprehension systems' robustness, revealing models' sensitivity to variations in expression (Jia & Liang, 2017). Similarly, by analyzing misclassified examples in NLI models, researchers have constructed stress test sets to determine whether models fail to reason correctly because certain samples do not fit specific patterns (Naik et al., 2018). Methods to enhance robustness have mostly originated from the data perspective, such as improving a text classification model's ability to handle different expressions through data augmentation techniques (Wei & Zou, 2019). Alternatively, lightweight adversarial filtering methods have been employed to filter biases from the original datasets (Le Bras et al., 2020).

With the advent of LLMs, the concept of "preference" has become more pronounced. Researchers have found that differently phrased prompts can lead to vastly different generated results, prompting the development of various prompt-improvement strategies. For example, some methods involve formulating multiple candidate instructions and using reinforcement learning to select the best prompt, while adjusting it in real-time according to the context (Zhou et al., 2022). Other approaches involve adding slight directional stimuli to the prompt to guide LLMs toward producing more optimal

outputs (Li et al., 2024). Rewriting queries to bridge the gap between search inputs and the required knowledge, thereby improving retrieval performance (Ma et al., 2023).

# 3 METHODS

## 3.1 TASK FORMULATION

Text classification is a fundamental task in natural language processing that involves assigning predefined categories or labels to textual inputs. The task can be formally defined as follows: Given a dataset $D = (x_i, y_i)_{i=0}^N$, where each $x_i$ represents the input text or document, and $y_i$ denotes the corresponding class label or category. The goal of text classification is to learn a model that maps each input text $x_i$ to its appropriate label $y_i$.

## 3.2 FRAMEWORK

As shown in Figure 2, We propose a framework for leveraging an LLM to rewrite text classification inputs, including training phrase (a&b) and inference phrase (c).

For the training phrase, we separated a portion of the dataset and utilized the LLM to paraphrase this data. We generated multiple paraphrased variations **(a.1)**. Next, we employed the Classifier to perform inference on the paraphrased data. From the results, we selected certain paraphrases to be used as fine-tuning data for the LLM **(a.2)**. Following this, we fine-tuned the LLM **(a.3)** and used it to paraphrase the original data once more, resulting in a new set of paraphrased samples **(a.4)**. We then repeated the inference process with the Classifier to identify suitable data for Direct Preference Optimization (DPO) training for the LLM **(a.5)**. Subsequently, we carried out DPO training on the LLM **(a.6)**. The DPO training data was then input into the Classifier to retrieve embedding data for these paraphrased samples **(b.1)**. Using this embedding data, we trained a filter **(b.2)**.

The mathematical formulation is presented below, where $\mathcal{P}$ denotes the large language model (LLM) paraphraser and $\mathcal{C}'$ represents the selector based on the classifier $\mathcal{C}$'s judgments (specific methods are described in the DPO-training section).

$$\forall (x_i, y_i) \in \mathcal{D}, \ \mathcal{P}(x_i) = \{p_{i1}, \ldots, p_{ik}\}$$

$$\mathcal{D}_{\text{ft}} = \bigcup_{i=1}^N \{p_{ij} \mid \mathcal{C}(p_{ij}) = y_i\}$$

$$\mathcal{D}_{\text{dpo}} = \bigcup_{i=1}^N \left\{ (p_{ij}^+, p_{ik}^-) \ \middle| \ \begin{matrix} p_{ij}^+ \in \mathcal{P}(x_i), \ \mathcal{C}'(p_{ij}^+) = 1, \\ p_{ik}^- \in \mathcal{P}(x_i), \ \mathcal{C}'(p_{ik}^-) = 0 \end{matrix} \right\}$$

For the inference phrase, we integrated the workflows of both the LLM and Classifier with the filter, establishing a comprehensive process: Utilizing an LLM, multiple sampling-based rewrites of the input are generated **(c.1)**. Subsequently, a filter is employed to discard low-quality rewrites **(c.2)**. The high-quality rewrites, along with the original input, are then processed using a Classifier for inference **(c.3)**. The output with the highest confidence is selected as the final prediction.

## 3.3 LLM TRAINING

**SFT** Firstly, we perform a supervised fine-tuning (SFT) process on the LLM. This fine-tuning primarily aims to stabilize its output format and provide a preliminary warm-up for preference modeling. We input samples from the dataset into the LLM and, based on feedback from the Classifier, separate out the well-rewritten outputs. These selected outputs are then fed back into the LLM for fine-tuning. This process resembles a self-boosting training mechanism for the LLM, enabling it to gain an initial understanding of the kind of rewrites it should produce. A well-rewritten output selection rule here is to make the output distribution of the Classifier closer to the correct label.

**DPO** Next, we employ the DPO (Direct Preference Optimization) approach to conduct preference optimization training for the LLM. In this optimization process, unlike previous methods that fit

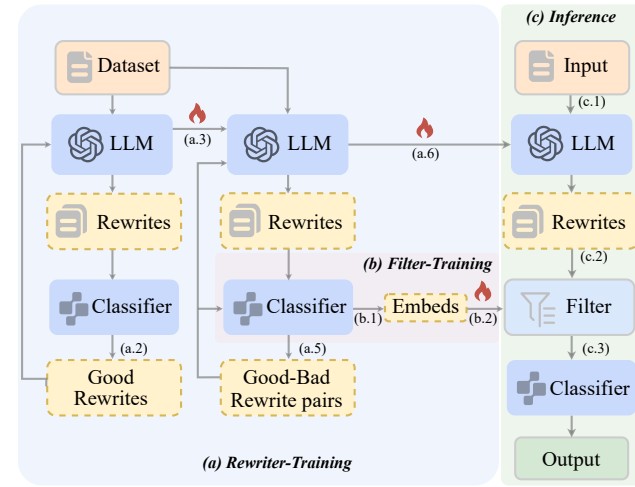

Figure 2: Overall architecture of our approach. (a) The process of obtaining, and evaluating the paraphrased data, and using them to train the model. (b) Train a filter within the DPO training data's encodings of the Classifier. (c) The final inference process.

human preferences, the Classifier is tasked with providing preference data to the LLM. Specifically, for the same input, if the rewritten version generated by the LLM shifts the predictions of the Classifier in the correct direction, this rewrite is considered a positive response; otherwise, it is deemed a negative response. The data organization method is as follows:

- First, we utilize the Classifier's prediction correctness and confidence level (from softmax results) as filtering criteria.

- For correctly predicted original inputs: rewrites that maintain correctness and improve confidence are considered positive; others negative.

- For incorrectly predicted inputs: rewrites that either correct the prediction or reduce confidence (while remaining incorrect) are positive; others negative.

- When only positive candidates exist, the original input is treated as negative; when no positives exist, the original is considered positive.

In summary, while the fine-tuning part equips the LLM with an understanding of how to generate sentences with "semantically similar" meanings, the DPO training phase teaches the LLM the preferences of the Classifier. These preferences highlight that two sentences with similar semantics can elicit varying levels of reasoning ability from the Classifier due to differences in vocabulary, morphology, or syntax. By pairing these sentences as inputs for training the LLM, we can leverage its extensive knowledge of natural language to capture subtle distinctions that are challenging to identify using rule-based methods and reformulate sentences into forms that the Classifier is more adept at handling.

## 3.4 INFERENCE

After obtaining a trained rewriter, during the inference phase, we initially filter the input data. Samples with a confidence level above a certain threshold $\alpha$ are deemed unnecessary for rewriting enhancement. For samples with a confidence level below $\alpha$, we employ the rewriter to generate rewritten versions using prompts consistent with those used during training. We sample $K$ times to obtain $K$ rewritten texts. For binary classification tasks, this involves rewriting both sentences and combining them with the other original text, resulting in $2 * K$ rewritten texts. These rewritten samples are then screened using a filter to retain those considered superior to the original text. Finally, the small model performs inference using these rewritten texts and the original text, selecting the output with the highest confidence as the predicted result.

**Filter**   Due to the sampling strategy of the LLM, it is challenging to ascertain what specific preferences they have learned from Classifiers. Additionally, due to the enormous number of parameters in the LLM, it is challenging to analyze the extent to which it models preferences for the Classifier. To evaluate the outputs at a data level, we train a Filter using the DPO data from the previous step to perform filtering. One crucial component is the filter, which we need to train effectively to help us eliminate low-quality rewrites. We employ the following approach to train the filter:

Based on existing research on pre-trained models with a transformer (Vaswani, 2017) architecture, such as BERT or RoBERTa, different layers in these models focus on different types of information. Word or phrase information tends to concentrate in the lower layers, syntactic and grammatical information in the middle layers, and this information becomes diluted in the higher layers, which focus more on semantics (Jawahar et al., 2019). In semantic text classification tasks, semantics play a crucial role in the final decision-making while preferences should manifest in different expressions; hence, these preferences are likely to be reflected in the mid to lower layers' encodings.

Due to the minimal differences between the original and rewritten texts, it is challenging to directly capture the Classifier's preference information from the text. To deeply model the Classifier's preferences, we utilize its internal encoding information. We use the embeddings of the token, which is employed for the final classification, across all layers in the model. For RoBERTa, specifically, we use the text's [CLS] (Devlin, 2018) embeddings from each layer of the model as a substitute for directly using the original text. We concatenate the [CLS] embeddings of the positive sample from all layers and then subtract the embeddings of the negative sample to obtain a difference embedding, labeled as "true". For the other half of the data, reverse the operation and label the result as "false". These difference embeddings are then passed through several fully connected layers (Rumelhart et al., 1986) to perform a binary classification task. Taking Roberta-Large as an example, the process of filter training is as follows:

$$\mathbf{H}(x) = \bigoplus_{l=1}^{L} \mathbf{h}^l(x), \mathbf{h}^l(x) = \text{RoBERTa}^{(l,0)}(x) \tag{1}$$

$$\Delta(x^+, x^-) = \begin{cases} \mathbf{H}(x^+) - \mathbf{H}(x^-) & y = 1 \\ \mathbf{H}(x^-) - \mathbf{H}(x^+) & y = 0 \end{cases} \tag{2}$$

$$f_\theta(\Delta) = FFNN(\Delta(x^+, x^-)) \tag{3}$$

$$\tag{4}$$

For the training loss of the filter:

$$\mathcal{L}(\theta) = -\frac{1}{N} \sum_{i=1}^{N} \Big[ y_i \log \hat{y}_i + (1 - y_i) \log\big(1 - \hat{y}_i\big) \Big] \tag{5}$$

where $\hat{y}_i = sigmoid(f_\theta(\Delta_i))$ is the predicted score of the classifier.

The results indicate that this method is effective, confirming that the Classifier's preferences are indeed reflected in its internal parameters. With this filter in place, we are able to filter the LLM's paraphrased outputs to achieve higher-quality final results.

## 4   EXPERIMENTS

### 4.1   EXPERIMENT SETTINGS

**Benchmarks**   We focus on semantic classification tasks with ample training data, which allows us to collect preference data for Classifiers. Our experiments are conducted on a widely-used benchmark GLUE (Wang, 2018), which consists of multiple subsets. Among these subsets, the CoLA task is grammar-based; the remaining tasks are semantic in nature. As samples with grammar errors are automatically corrected by LLMs during rewriting, this renders our approach ineffective in those cases. Therefore, we utilize the subsets excluding CoLA. Our objective is to examine the ability that our work can enhance performance across various semantic classification tasks by assisting Classifiers in rewriting inputs. For cost considerations, in subsequent analytical experiments, we conducted experiments on three subsets: MRPC, MNLI, and SST-2.

Table 2: Experimental results on GLUE development set. The performance of the text complexity reduction method is represented by the average of the four metrics mentioned above. We use accuracy as the evaluation metric.

| Methods | MNLI | SST-2 | QNLI | MRPC | QQP | RTE | Avg. |
|---|---|---|---|---|---|---|---|
| BART-base | 83.99 | 94.40 | 91.82 | 85.33 | 89.97 | 79.59 | 87.52 |
| +Complexity reduction | 83.84 | 94.66 | 91.67 | 85.51 | 89.93 | 80.27 | 87.65 |
| +Distill-R1-Qwen-1.5B | 84.49 | 94.66 | 92.02 | 86.11 | 90.03 | **81.99** | 88.22 |
| +LLaMA-2-13B-chat | 84.54 | 94.91 | 92.02 | **86.26** | 90.03 | 80.95 | 88.12 |
| +Qwen-2.5-14B-Instruct | **84.68** | **95.16** | **92.07** | 85.86 | **90.05** | 81.63 | **88.24** |
| RoBERTa-Large | 88.72 | 96.20 | 93.82 | 83.71 | 90.90 | 86.19 | 89.92 |
| +Complexity reduction | 88.67 | 96.39 | 93.38 | 86.03 | 90.30 | 83.47 | 89.71 |
| +Distill-R1-Qwen-1.5B | 89.12 | 96.69 | 93.92 | 86.52 | 91.08 | 86.87 | 90.70 |
| +LLaMA-2-13B-chat | **89.81** | 96.46 | 93.97 | 87.13 | **91.10** | **87.55** | **91.00** |
| +Qwen-2.5-14B-Instruct | 89.46 | **96.95** | **93.98** | **87.77** | 90.95 | 86.87 | **91.00** |
| Qwen-2.5-7B-Instruct | 66.06 | 90.58 | 78.87 | 66.19 | 70.04 | 79.59 | 75.22 |
| +Complexity reduction | 66.73 | 90.33 | 78.17 | 66.67 | 71.43 | 76.88 | 75.04 |
| +Distill-R1-Qwen-1.5B | 68.25 | 90.84 | 80.73 | 66.85 | 72.12 | 80.27 | 76.51 |
| +LLaMA-2-13B-chat | **68.67** | 91.09 | 80.89 | 67.61 | 73.21 | 80.95 | 77.07 |
| +Qwen-2.5-14B-Instruct | 68.34 | **91.86** | **82.94** | **68.67** | **74.20** | **81.63** | **77.94** |

**Large Models and Classifiers** For the model rewriting task, we utilize LLaMA-2-13B-Chat (Touvron et al., 2023), Qwen-2.5-14B-Instruct (Yang et al., 2024) and DeepSeek-R1-Distill-Qwen-1.5B (Guo et al., 2025). For the GLUE dataset, we employ BART-base (Lewis, 2019) and RoBERTa-Large (Liu, 2019) as expert Classifiers, with each model trained on the different subsets of GLUE and Qwen2.5-7B-Instruct as a zero-shot Classifier. Due to cost considerations, only RoBERTa-Large and LLaMA-13B-Chat were used in the analytical experiments. Additionally, Since DeepSeek-R1-Distill-Qwen-1.5B is significantly slower at generating training data, we primarily used data produced by Qwen-2.5-14B-Instruct (with <think></think> prepended to the output to adapt it for reasoning-model training), mixed with a small portion of DeepSeek-R1-Distill-Qwen-1.5B's own generated data containing reasoning processes, to train the model.

## 4.2 MAIN RESULTS

As shown in Table 2, in contrast to the straightforward method of lowering the complexity of input text, our method effectively enhances performance across various subsets of the GLUE benchmark when applied to three different Classifiers. Specifically, it boosts the performance of BART-base by 0.72 points and RoBERTa-large by 1.08 points on the overall GLUE dataset. For BART, our method yields relatively uniform improvements across various subsets. In contrast, for RoBERTa, the rewriting approach shows particularly significant enhancement on the MRPC dataset. We hypothesize that this may be due to the presence of considerable noise in the MRPC data for RoBERTa, which is somewhat mitigated during the rewriting process by the LLM. For the Qwen model, since its task accuracy under zero-shot conditions is significantly lower, our method demonstrates a more pronounced improvement, achieving an increase of 2.72 points. Additionally, it is worth noting that on tasks like GLUE, large models such as LLaMA and Qwen inherently underperform compared to models like BART and RoBERTa, which are trained on full datasets. However, even in such cases, our method can still enable them to surpass the capabilities of models that originally outperformed them. In addition, our approach does not degrade the original performance of the Classifiers on any of the datasets. This demonstrates that our method is not only effective in improving accuracy but also safe to use without risking negative impacts on the models' inherent capabilities.

## 4.3 ABLATION STUDY

We conducted ablation experiments to investigate the roles of different components in our work. We broke down the entire framework into three parts and examined their performance when removed: (a)

Table 3: Experimental results of the ablation study. Tested the effects of removing supervised fine-tuning, DPO training, and filtering separately.

| Methods | MNLI | MRPC | SST-2 |
|---|---|---|---|
| RoBERTa-Large | 88.72 | 83.71 | 96.20 |
| Our Method | **89.81** | **87.13** | **96.46** |
| w/o Filter | 89.51 | 86.89 | 96.20 |
| w/o DPO | 89.31 | 86.37 | 95.75 |
| w/o SFT | 88.76 | 85.10 | 95.50 |
| w Reverse Training | 87.82 | 83.83 | 95.18 |

the fine-tuning process of the LLM, (b) the DPO training process, and (c) the filter screening process. The results are presented in Table 3.

(1) First, we attempted to eliminate the filter and relied solely on the Classifier's own confidence to determine which samples needed rewriting. We used all rewritten versions as well as the original output, directly selecting the result with the highest confidence score as the final output. In this scenario, performance declined, which we attribute to the Classifier's confidence not fully reflecting correctness. Furthermore, the proportion of examples that could be correctly predicted by the Classifier was very high, so even a small fraction being incorrectly rewritten by the LLM significantly impacted the overall outcome.

(2) We verified that not using DPO resulted in a more significant performance drop compared to not using the filter. We believe this is because the fine-tuning process alone allows the LLM to learn only what constitutes a good rewrite without a direct understanding of bad rewrites, leading to suboptimal overall rewriting quality. This also demonstrates that our DPO training process successfully modeled the Classifier's preferences through the LLM and taught the LLM how to cater to these preferences.

(3) We tried performing DPO without fine-tuning the model first. Since the DPO training process focuses on contrasting good and bad outcomes, the specific output format and content of the rewriting task were not adequately trained. In this case, the LLM tended to generate a lot of unnecessary additional content. Even with truncation and screening, the final performance decreased significantly. Therefore, it is crucial to use a small amount of data to fine-tune the model to stabilize the output format.

(4) We attempted to train the rewriter using opposite data, specifically utilizing data that should have been classified as poor rewrites. This approach serves to validate, from a reverse perspective, that we have successfully captured the model's preferences. Contrary to our expectations, for the MRPC dataset, even a rewriter fitted to its negative preferences ultimately resulted in some performance improvement. The reason for this may align with the analysis we provided in Section 4.2.

## 4.4 REWRITING ANALYSIS

We begin by comparing our reinforcement learning-based rewriting approach with methods that rely on manually designed prompts. As shown in Table 4, While the hand-crafted prompt approach demonstrates certain effectiveness in specific dataset-model combinations, its performance exhibits considerable instability across different datasets. This inconsistency highlights a fundamental challenge: predefined prompts struggle to generalize appropriate rewriting directions that reliably align with the preferences of the downstream classifier. It is precisely this inability to consistently derive optimal rewriting strategies through prompt engineering that motivates our transition to reinforcement learning. By explicitly modeling the task model's preferences via SFT+DPO, our approach achieves stable and effective text rewriting, leading to more robust performance improvements across diverse datasets.

We analyzed the proportion of samples rewritten in our method and the proportion of rewritten results used as final input. We only used LLM rewriting for some of the samples whose confidence is below the set threshold. The rule for selecting the final input involves a filtering out rewrites it deems inferior to the original input. The version either the original or rewritten that results in the highest confidence

Table 4: Experimental results comparing our method with approaches using manually designed prompts. Among these, DAR stands for "Disambiguation and Reference Resolution", LSR for "Logical Structure Refinement", and FCR for "Formal Coherence Reformulation". The rewriter is Qwen-2.5-14B-Instruct. For specific prompt details, please refer to the appendix.

| Methods | MNLI | SST-2 | QNLI | MRPC | QQP | RTE | Avg. |
|---|---|---|---|---|---|---|---|
| BART-base | 83.99 | 94.40 | 91.82 | 85.33 | 89.97 | 79.59 | 87.52 |
| +Prompt-DAR | 84.04 | 94.40 | 91.92 | 85.16 | 89.95 | 80.27 | 87.62 |
| +Prompt-LSR | 84.59 | 94.91 | 91.77 | 85.45 | 89.80 | 80.27 | 87.80 |
| +Prompt-FCR | 84.09 | 94.66 | 91.97 | 85.04 | 90.00 | 80.27 | 87.67 |
| +Our method | **84.68** | **95.16** | **92.07** | **85.86** | **90.05** | **81.63** | **88.24** |
| RoBERTa-Large | 88.72 | 96.20 | 93.82 | 83.71 | 90.90 | 86.19 | 89.92 |
| +Prompt-DAR | 89.36 | 96.20 | 92.72 | 86.26 | 90.80 | 86.87 | 90.37 |
| +Prompt-LSR | 89.26 | 95.96 | 93.88 | 86.72 | 90.75 | 82.79 | 89.89 |
| +Prompt-FCR | 89.31 | 96.20 | 93.78 | 85.10 | **90.95** | 84.15 | 89.92 |
| +Our method | **89.46** | **96.95** | **93.98** | **87.77** | **90.95** | **86.87** | **91.00** |
| Qwen-2.5-7B-Instruct | 66.06 | 90.58 | 78.87 | 66.19 | 70.04 | 79.59 | 75.22 |
| +Prompt-DAR | 66.11 | 90.33 | 78.82 | 65.89 | 69.90 | 79.59 | 75.11 |
| +Prompt-LSR | 66.11 | 90.32 | 79.12 | 66.65 | 69.94 | 80.27 | 75.40 |
| +Prompt-FCR | 66.21 | 91.09 | 78.87 | 66.37 | 70.04 | 80.95 | 75.59 |
| +Our method | **68.34** | **91.86** | **82.94** | **68.67** | **74.20** | **81.63** | **77.94** |

Table 5: Experimental results of the proportion of rewriting results used. %Rewritten reflects how many samples were rewritten, %Utilization reflects how many samples used the rewrite as the final input, while %Right2Wrong and %Wrong2Right indicate the changes in the Classifier's prediction results for samples in %Utilization.

| Methods | MNLI | MRPC | SST-2 |
|---|---|---|---|
| %Rewritten | 12.40% | 7.42% | 0.52% |
| %Utilization | 10.03% | 6.40% | 0.26% |
| %Right2Wrong | 11.96% | 13.64% | 0% |
| %Wrong2Right | 22.83% | 67.05% | 100% |

output from the Classifier is chosen. As shown in Table 5, on the MNLI dataset, approximately 10% of the samples are replaced with rewritten results. Of these, the majority do not alter the Classifier's original predictions, about 22% correct previously incorrect predictions, and about 12% lead to incorrect predictions. On the MRPC dataset, although fewer samples use the rewritten outputs, the correction rate reaches 67%, which explains our strong performance on MRPC. For the SST-2 dataset, since the Classifier's accuracy is already over 96% with highly confident outputs, only 0.26% of the samples are deemed to require the rewritten version after filtering. Notably, all of these cases involve correcting errors made by the Classifier when using the rewritten input.

## 5 CONCLUSION

In this paper, we propose a reinforcement learning-based method for modeling task model preferences, which leverages LLMs to adaptively rewrite input texts, thereby unleashing the full performance potential of task models. It is important to emphasize that our work is not merely aimed at achieving improvements on experimental benchmarks, but rather focuses on unlocking the upper capability boundaries of task models through preference-guided input rewriting. Our approach provides a general and flexible framework that enhances model performance without modifying the underlying architecture, offering new insights into model interaction and capability expansion.

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

# A APPENDIX

## A.1 GENERALIZATION ANALYSIS

As shown in Table 6, we experiment with utilizing an LLM rewriter, trained on a specific dataset, to enhance performance on other datasets without employing a filter. The experimental results indicate that for the MRPC and MNLI datasets, all three rewriters contribute to improvements in the performance of the Classifier. However, on the relatively simpler SST-2 dataset, even the SST-2-specific rewriter fails to enhance the performance without the filter. We hypothesize that these observations can be attributed to several factors. Firstly, LLMs inherently possess certain natural language knowledge, which allows them to assist Classifiers by rewriting inputs, even in the absence of deep cognition of specific preferences. Secondly, our approach primarily aims to enable Classifiers to better utilize their inherent capabilities. Given the simplicity of the SST-2 task, the abilities of the Classifier may already be maximized, making further improvements challenging. Lastly, we believe that the preferences of the Classifiers trained on these datasets share common traits. This may stem from the similarity of the tasks or the similarity of the model architectures.

## A.2 NUMBERS OF REWRITES ANALYSIS

Then, we investigated the impact of the number of rewrites per input sample on performance, as illustrated in Figure 3. Except for the SST-2 dataset, we found that providing just one rewrite per sample is sufficient to improve the accuracy. In the case of SST-2, due to the already high accuracy of the Classifier, at least two rewrites per sample are needed to achieve noticeable performance gains. Furthermore, we observed that when the number of rewrites reaches three, the performance tends to converge. Beyond this point, additional rewrites may sometimes mislead the Classifier into making incorrect predictions with high confidence. Therefore, for our main experiments, we set the number of rewrites to three. This approach balances the benefits of rewrites with the risk of introducing errors.

## A.3 MANUAL PROMPT DETAILS

In order to systematically guide the language model to generate text rewrites with different stylistic and structural properties, we manually designed three distinct prompts, each targeting a specific aspect of textual improvement: Disambiguation and Reference Resolution (DAR), Logical Structure Refinement (LSR), and Formal Coherence Restoration (FCR). The detailed content of each prompt is provided in Table 7.

Table 6: Experimental results of the generalization study. All methods are without a filter.

| Methods | MNLI | MRPC | SST-2 |
|---|---|---|---|
| RoBERTa-Large | 88.72 | 83.71 | 96.20 |
| + MNLI-Rewriter | **89.51** | 86.61 | 95.95 |
| + MRPC-Rewriter | 88.97 | **86.89** | 95.69 |
| + SST-2-Rewriter | 88.82 | 86.55 | **96.20** |

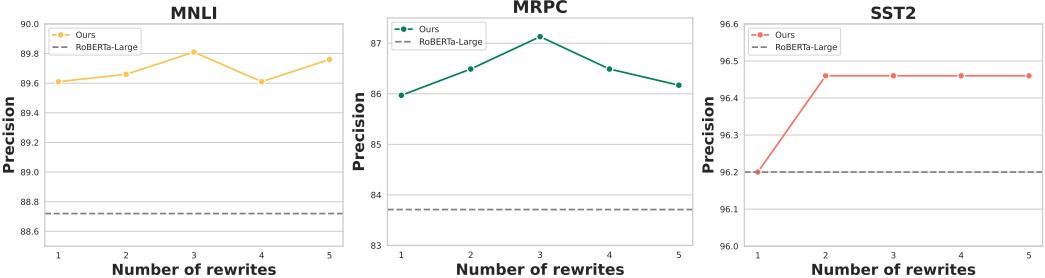

Figure 3: Effect of the number of rewriting.

### A.4 PARAMETER SETTINGS

In our experiments, the threshold $\alpha$ is set to 0.6, the sample times K is set to 3, the learning rate and training epochs for SFT of the LLM are set to 1e-4 and 3, the learning rate and training epochs for DPO of the LLM are set to 5e-6 and 3, and the temperature during inference is set to 0.7.

Notably, if we were to sample the classifier's output confidence and accuracy and set $\alpha$ accordingly, for instance, for a model with 85% accuracy, the optimal $\alpha$ should be such that samples with confidence below $\alpha$ account for 15% of the total, the performance could be further improved. However, for the sake of generalization, we did not adopt this approach in our experiments. This implies that the actual upper bound of our method is slightly higher than what is demonstrated in the main experiments.

### A.5 USE OF LARGE LANGUAGE MODELS

During the preparation of this work, we used Large Language Models (LLMs) solely for the purpose of improving language and clarity. Specifically, LLMs were used for proofreading, grammar correction, and minor phrasing improvements.

We reviewed and edited all output generated by the LLMs, and takes full responsibility for all content and ideas presented in this work.

Table 7: Detailed prompt designs for each rewriting strategy.

| Name | Prompt Content |
| --- | --- |
| DAR | Rewrite the following text to resolve any ambiguity and ensure clear reference resolution. Requirements:
1. Identify and clarify all ambiguous pronouns (e.g., 'it', 'this', 'they') by replacing them with the specific nouns they refer to.
2. Disambiguate any words or phrases that could have multiple meanings.
3. Ensure that every reference is unmistakably clear.
4. Maintain the core semantics of the original text unchanged.
5. Output only the rewritten text.
Text: {text}
Rewritten text: |
| LSR | Rewrite the following text to make its logic more clear and direct. Requirements:
1. Express implied relationships (e.g., causality, contrast) using explicit linking words (e.g., 'because', 'therefore', 'but').
2. Simplify complex clause structures, striving to use simple subject-verb-object sentences.
3. Absolutely maintain the core semantics of the original text unchanged.
4. Just output rewrited text. Do not output any extra content.
Text: {text}
Rewritten text: |
| FCR | Rewrite the following text to enhance its formal coherence and semantic precision. Requirements:
1. Replace colloquial expressions and informal phrasing with their formal equivalents.
2. Ensure strict grammatical correctness and syntactic completeness.
3. Maintain precise semantic equivalence while improving textual fluency.
4. Resolve any ellipsis or fragmented structures into complete sentences.
5. Just output rewrited text. Do not output any extra content.
Text: {text}
Rewritten text: |

