# OpenReview forum: "Bridging the Preference Gap: Post-Training Input Rewriting with Large Language Models"
_ICLR.cc/2026/Conference — ICLR 2026 Conference Withdrawn Submission_

### Official Review · Reviewer_53au · 2025-10-15

**Soundness:** 2
**Presentation:** 3
**Contribution:** 1
**Rating:** 2
**Confidence:** 4

**Summary:**

This paper proposes a post-training input rewriting framework that leverages large language models (LLMs) to rewrite input texts at inference time, aligning them with the “preferences” of a fixed downstream classifier (e.g., RoBERTa, BART) to boost its performance on GLUE. The method involves: (1) automatically constructing preference-aligned training data, (2) fine-tuning the LLM via SFT and DPO, and (3) applying a multi-sampling + classifier-embedding-based filtering strategy during inference.

**Strengths:**

- The experimental pipeline is relatively complete, including ablation studies, generalization analysis, and comparisons with hand-crafted prompts.
- It presents a post-hoc, inference-time approach that avoids retraining the classifier.
- The paper empirically challenges the common assumption that reducing textual complexity improves model performance.

**Weaknesses:**

- **Motivation is weak and conceptually muddled**: The term “preference” is used loosely without clear definition or causal analysis. The paper conflates model limitations (e.g., poor generalization to paraphrases) with inherent “preferences,” then proposes a complex workaround instead of addressing root causes.
- **Performance gains are marginal**: The method improves RoBERTa-Large by only **+1.08** GLUE points and BART-base by **+0.72**. The large gain on MRPC (+3.4) is isolated and likely stems from dataset noise rather than a robust mechanism, as noted by the authors themselves.
- **Excessive engineering complexity**: The pipeline requires SFT + DPO training of an LLM, construction of a separate classifier-embedding-based filter, and multi-sample inference—yet yields sub-1% gains. This makes the approach impractical for real-world deployment.
- **Missing key baselines**: No comparison with standard test-time augmentation (TTA), self-consistency decoding, or even simple prompt ensembling—methods that are far cheaper and often equally effective.
- **Limited scientific insight**: The work offers no analysis of what linguistic features the classifier actually “prefers” or how the LLM learns them. It remains a black-box performance patch with little theoretical or practical generalizability.

**Questions:**

1. How do you formally define “model preference”? Can you provide evidence that it is a stable, intrinsic property of the classifier—not an artifact of training data bias or insufficient robustness?
2. Why not compare against standard test-time augmentation or ensemble-based inference, which are simpler and more widely adopted?
3. Are the reported gains (e.g., +1.08 on GLUE for RoBERTa) statistically significant? Were multiple random seeds or runs used to rule out variance?
4. Given the high computational and latency overhead of your pipeline (LLM rewriting + filtering), how do you justify its practical utility over simply fine-tuning the classifier further or using a stronger base model?

---

### Official Review · Reviewer_7n88 · 2025-10-31

**Soundness:** 2
**Presentation:** 3
**Contribution:** 2
**Rating:** 4
**Confidence:** 3

**Summary:**

This work propose a framework that trains an LLM to rewrite input texts to assist a classifier to make better classification in semantic classification tasks according to its preference. The LLM rewriter is trained via SFT and DPO with data labeled by the classifier to align with the classifier's preference. A filter module is also trained using the classifer's embeds to filter inferior rewrites from the LLM rewriter's multiple samplings. Experiments show that the framework helps improve the performance of bert-based classifiers and LLMs on GLUE benchmark.

**Strengths:**

1. This paper propose a framework that leverage classifier's inherent preference to improve its accuracy rather than forcefully correct the preference, providing a new prospective for enhancing the performance of classifiers.
2. The training process is concise and easy to conduct, which enhances the applicability of the framework.

**Weaknesses:**

1. The experiment lacks a comparison with the effect of directly fine-tuning the classifier with diverse rewriting formats, which is necessary to prove that leveraging the preference of classifier is better than directly correcting it.
2. The research field of this work is limited. This work focuses only on the semantic classification tasks, and GLUE is a relatively simple benchmark for current models. Since the authors said their work "focuses on unlocking the upper capability boundaries of task models through preference-guided input rewriting" in the conclusion, it is necessary to validate this method on more tasks, such as instruction following.
3. The base models selected in the experiments are out-dated. Stronger baselines should be taken into account, such as reasoning models like Qwen3-8B and other LLM-based classifiers.

**Questions:**

See weaknesses.

---

### Official Review · Reviewer_ShwC · 2025-11-01

**Soundness:** 2
**Presentation:** 2
**Contribution:** 1
**Rating:** 2
**Confidence:** 5

**Summary:**

This paper presents a framework that employs large language models (LLMs) to rewrite input texts to align with the preferences of a target classifier. The authors propose a training paradigm for the LLM, accompanied by an automated data construction pipeline that encapsulates classifier-specific characteristics. A multi-sampling and filtering strategy is further introduced to mitigate the inherent instability of LLM-generated outputs. Empirical evaluations on semantic classification datasets demonstrate that the proposed framework yields improvements in classifier performance.

**Strengths:**

1. This paper empirically validates the distinction of model preferences from human linguistic cognition, demonstrating that traditional text complexity metrics, such as sentence length and lexical rarity, cannot reliably predict model behavior.
2. This paper proposes a method to train LLMs to capture the preferences of the Classifier, along with a technique for the automated construction of training data.
3. This paper presents a method that combines multiple sampling with a filtering strategy to address the issue of instability in the outputs generated by the LLM.

**Weaknesses:**

1. The practicality of the approach appears limited. In the title of the paper, it seems that the authors propose an empirical paradigm; however, the scope of the study is restricted to text classification tasks, specifically the MRPC, MNLI, and SST-2 datasets. These benchmarks have already achieved near-saturated performance (e.g., 96.20% accuracy with RoBERTa-Large on SST-2). Hence, doubts remain about the meaningfulness of the work, and it is unclear whether the proposed framework can generalize to more diverse and complex scenarios, such as text generation or reasoning tasks.
2. Although the authors claim that "*Empirical evaluations on semantic classification datasets demonstrate that our framework **significantly** improves classifiers' performances,*" the reported performance gains are relatively small and may fall within the range of random variation (e.g., +0.45% on QNLI with BART-Base). To substantiate the claim of significant improvement, multiple experimental runs and statistical significance tests should be conducted. Moreover, the efficiency of the proposed approach is questionable, as the framework requires multiple sampling runs to obtain the final results, which could substantially increase computational overhead.
3. The experimental evaluation lacks an ablation study or a detailed analysis of the proposed filtering module. It remains unclear how much the filtering component performs and how it compares to the conventional RoBERTa classification. A more comprehensive examination of this module would strengthen the empirical validity of the work.

**Questions:**

See "Weaknesses."

---

### Official Review · Reviewer_oBL7 · 2025-11-01

**Soundness:** 2
**Presentation:** 3
**Contribution:** 2
**Rating:** 4
**Confidence:** 3

**Summary:**

The paper proposes a post-training framework that uses LLMs to rewrite classification inputs according to a classifier’s inherent preferences. It includes SFT, DPO, and a filtering stage based on classifier embeddings. Experiments on GLUE show small but consistent gains.

**Strengths:**

1. The shift from "eliminating preferences" to "adapting to preferences" is interesting.
2. The three-stage training from SFT, DPO to Filter, is technically sound in isolation.
3. In the current experiments, the effectiveness on the selected baselines and benchmarks is demonstrated, and through ablations, the authors effectively show the necessity of each component.

**Weaknesses:**

1. The approach is conceptually interesting but not a strict or theoretically grounded post-training method. Training stability and convergence are not statistically validated.

2. The selection of datasets and baselines is limited, lacking generalization analysis.

3. Empirical improvements are minor; significance not verified.

4. The observed performance improvements might stem from the LLM memorizing task-specific linguistic patterns rather than capturing genuine preference alignment. I suggest adding evaluations on out-of-domain datasets to verify the robustness  of the proposed method.

4. Minor presentation errors – e.g., (a.1) and (a.4) are mentioned but missing in Figure 2.

**Questions:**

1. How stable is the DPO-based training process? Have you observed consistent convergence across multiple random seeds?

2. Could the filter generalize across classifiers, or must it be retrained for each new target model?

---

### Note · Authors · 2025-11-20

I have read and agree with the venue's withdrawal policy on behalf of myself and my co-authors.